# Evolution of Suspected Cat Abuse Between 2020 and 2024 in the Community of Madrid (Spain)

**DOI:** 10.3390/ani15192892

**Published:** 2025-10-03

**Authors:** Nicolás Aradilla, Javier María De Pablo-Moreno, Néstor Porras, Blanca Chinchilla, Antonio Rodríguez-Bertos

**Affiliations:** 1VISAVET Health Surveillance Center, Complutense University of Madrid, 28040 Madrid, Spain; jdepab01@ucm.es (J.M.D.P.-M.); nestorpo@ucm.es (N.P.); bchinchi@ucm.es (B.C.); arbertos@ucm.es (A.R.-B.); 2Faculty of Veterinary Medicine, Complutense University of Madrid, 28040 Madrid, Spain; 3Department of Animal Production, Faculty of Veterinary Medicine, Complutense University of Madrid, 28040 Madrid, Spain; 4Department of Internal Medicine and Animal Surgery, Faculty of Veterinary Medicine, Complutense University of Madrid, 28040 Madrid, Spain

**Keywords:** veterinary forensics, cat abuse, forensic necropsy, non-natural death, feline colonies, pathology

## Abstract

**Simple Summary:**

Animal abuse is a serious and underreported problem, with cats, particularly those living in colonies, being frequent victims. In Spain, reliable data on feline abuse are limited. This study evaluated the cause, manner and mechanisms of death from 53 cats over five years at the Pathology and Forensic Veterinary Unit of the VISAVET Health Surveillance Center in Madrid (Spain). More than half of the cats died from non-natural causes, including blunt force trauma, poisoning, and projectile injuries, while the remaining cats died from natural causes. Compared to a previous regional study, suspected abuse cases have increased, including new instances of antifreeze poisoning. These findings provide insight into trends in feline abuse and underscore the need for improved detection, prevention, and professional awareness to protect vulnerable animal populations. Feline colonies continue to be a point of societal conflict, and forensic evidence suggests that they are a primary target for this type of abuse.

**Abstract:**

Despite the well-established link between animal abuse and interpersonal violence, animal mistreatment remains a global issue. Challenges such as inconsistent legislation, limited training of specialized pathologists, and lack of regional data persist and must be addressed. In Spain, the real incidence of animal abuse is unknown, and the literature on the topic remains scarce. This study aims to assess the evolution of feline abuse cases in the Community of Madrid, Spain, since the publication of a previous study in the same region. Over a five-year period (2020–2024), 53 cats with suspected abuse were diagnosed at the Pathology and Forensic Veterinary Unit of the VISAVET Health Surveillance Center. Thirty-one cats (58.49%) died from non-natural causes: 17 (32.07%) due to blunt force trauma, eight (15.09%) due to poisoning, and six (11.32%) due to projectile injuries. Natural causes accounted for 21 cases (39.62%). Compared to the previous study, data suggest a possible upward trend in the number of cats referred with suspected animal abuse, including eight new cases of ethylene glycol poisoning. The correlation between the type of suspected abuse and final diagnosis was very low. This article examines current challenges related to animal violence, with particular emphasis on feline colonies, and promotes awareness among both veterinary and non-veterinary professionals.

## 1. Introduction

Animal abuse is a complex and unacceptable phenomenon that must be legislated and prosecuted [1]. The well-established association between animal and interpersonal violence has already been extensively documented in the scientific literature [1,2,3,4,5,6], with animal abuse recognized as one of the main predictive factors [3,7,8]. Consequently, the accurate identification of violence towards animals is not only essential but also imperative to underscore the critical societal importance of establishing a precise diagnosis in each individual case. To the relevant and beneficial footprint left by One Health concept, One Welfare approach should also be adopted, recognizing the fundamental interconnections between animal welfare and human well-being [9]. The establishment of effective collaboration between human and veterinary forensics disciplines has been identified as a short-term goal to be achieved [10,11].

Animals suspected of abuse may be examined by either forensic veterinarians or general practitioners. Forensic veterinarians are primarily involved in legal investigations and the systematic documentation of evidence, whereas general practitioners often identify initial indicators of abuse and refer cases for specialized evaluation. Deceased animals suspected of abuse must undergo a rigorous necropsy procedure by a specialized pathologist in search of non-accidental injuries [5,12,13,14], to differentiate between cases of animal abuse and other accidental causes of death [15]. A complete study of the crime scene is highly valuable in this context [16], although it is not routinely conducted in most cases worldwide [5,12,17,18].

Despite the rapid and substantial growth that veterinary forensics have experienced in recent years [1,19,20,21], many non-forensic veterinary pathologists lack the necessary training to deal with these cases [22,23,24,25,26], and numerous challenges remain to be addressed [27]. Heterogeneous legislative frameworks and variable scientific development across countries [2,12] limit international communication, slowing down the advancement of the field. Veterinary clinicians will be confronted at some point in their careers with pet abuse cases [5,28], including physical, sexual, and emotional abuse, as well as neglect. Consequently, clinicians should be able to recognize the signs indicating animal abuse [6,29,30]. Experts in veterinary forensics not only face the responsibility of managing sensitive cases [1,29] under the accompanying judicial pressure [31], but also bear the duty of encouraging both veterinary and non-veterinary professionals to recognize cases of animal abuse and initiate appropriate forensic follow-up. Moreover, the presence of veterinarians in the public health sector who can accurately identify signs of animal abuse may help to restrain animal cruelty, and, in turn, prevent interpersonal violence [32].

The expression “cats are not small dogs” has already been used in the scientific literature [27,33], and considerable importance is now attributed to the fundamental differences between canines and felines in veterinary medicine. Nevertheless, within the field of veterinary forensics, the potential differences between cats and dogs in the context of animal abuse remain insufficiently investigated. Such distinctions need to be considered [27], just as human forensics accounts for differences among men, women and children. Assuming that sensitivity and predisposition to poisoning [12,34,35,36,37], ethological behavior [38], incidence of abuse [39], wound healing periods [40] or even decomposition processes [41] are identical between the two species would entail a substantial margin of error. Furthermore, extrapolating knowledge from human to veterinary forensics may cause comparative inaccuracies [42,43]. Species-specific studies are therefore essential, as they provide a foundation for more accurate and effective cross-comparisons in future research.

Feline colonies are defined by the new Spanish animal welfare law (Law 7/2023 of March 28, for the protection of the rights and welfare of animals) [44] as ¨a group of cats of the species *Felis catus*, living in a state of freedom or semi-freedom, which cannot be approached or kept easily by humans due to their low or no degree of socialization, but which develop their lives around them for their subsistence¨. Such colonies are indeed numerous worldwide. The global cat population was estimated at about 600 million in 2009 [45]. In the United States, there are an estimated 70 million free-roaming cats, almost the same as owned (86 million) [46]. In Spain, around 150,000 cats are abandoned each year [47]. Conflicts involving free-roaming cats from feline colonies frequently escalate to extreme measures such as trapping and culling, poisoning, or shooting, constituting a significant source of societal tension due to their close proximity and interaction with urban human populations [48]. As such, feline colonies represent a critical point in the investigation and prosecution of feline abuse and its potential association with interpersonal violence.

This law [44] also encompasses specific aspects of animal abuse, such as the definition of mistreatment as “any conduct, whether by action or omission, that causes pain, suffering, or injury to an animal and impairs its health, or leads to its death, when not legally justified”.

The upward trend in the number of veterinary forensic pathology articles published internationally [12,41,49,50], together with the legal frameworks being developed in many countries, has contributed to a marked increase in concern regarding animal abuse and welfare. Conversely, rising societal awareness also stimulates further research and the establishment of legislation, creating a mutually reinforcing cycle that leads to a greater number of animal mistreatment cases being brought to public attention [51]. Nevertheless, the real incidence of animal abuse in Spain remains unknown, with only a single article on the subject [32], underscoring the urgent need for further empirical research to accurately determine both the prevalence and societal impact on this form of violence.

To the authors’ knowledge, this is the second scientific article on animal abuse in Spain, and it represents the first research effort specifically focused on feline populations. In this context, the present retrospective study investigated the cause and manner of death in different cats suspected of animal abuse received at the Animal Pathology and Forensic Veterinary Unit of the VISAVET Health Surveillance Center at Complutense University of Madrid (Spain) between 2020 and 2024. Subsequently, a comparative discussion was performed with a similar study conducted at the same center between 2014 and 2019 [32], to assess trends in the prevalence of feline abuse in the Community of Madrid (Spain). These types of studies contribute to society by raising awareness of the importance of animal welfare and the current reality of animal abuse, thereby increasing sensitivity and the number of cases reported annually.

## 2. Materials and Methods

### 2.1. Cases Submitted and Postmortem Procedure

The retrospective study encompasses cases collected over a five-year period, ranging from 2020 to 2024. Forensic necropsies were performed on 53 cats suspected of having suffered non-natural deaths due to animal abuse. In all relevant cases, the chain of custody was preserved. The information provided to the pathologist for each case included the weight, breed, sex, age, microchip number identification, and the type of abuse suspected by the submitter. When available, information about previous pathologies or pharmacological treatments was also collected.

Cases were submitted by public institutions from the Community of Madrid (law enforcement agencies; *n* = 43), colony caretakers (*n* = 5), or owners of outdoor cats (*n* = 5), the majority of which originated from feline colonies (90.56%). Five cats (9.44%) were owned animals with outdoor access or indoor cats that died under circumstances suggestive of animal abuse occurring away from their homes.

All animals underwent the center’s standard protocol for forensic cases, including a complete and systematic necropsy, breed confirmation, microchip reading (when presented), determination of age and sex when unknown, and a histopathological evaluation of all organs to reach a definitive diagnosis. Age was assessed according to standard guidelines used by Rebollada-Merino (2020) [32] and established by other authors [52,53], based on tooth eruption and dental wear. Animals were classified as kitten (less than 6 months old), young (from 6 months to 3 years old), or adult (older than 3 years old). The postmortem interval was estimated by evaluating gross changes in the carcass, and each animal was assigned to one of the following five intervals: fresh (less than 3 days), bloat (3–10 days), active decay (10–20 days), advanced decay (20 days to 2 months), and dry remains or mummified (more than 2 months) [54]. Based on the pathologists’ criteria, photographs were systematically taken, and samples of all organs were collected and immersed in a 10% buffered formaldehyde solution. In cases with suspected poisoning and/or macroscopic compatible lesions, samples of target tissues were taken and frozen for further toxicological studies. A macroscopic and histopathological report was detailed after each complete process.

### 2.2. Histological Processing

After proper tissue fixation, standard processing of the samples was performed, including carving, tissue dehydration (Citadel 2000 Tissue Processor, Thermo Fisher Scientific, Waltham, MA, USA), paraffin block formation (HistoStar Embedding Workstation, Thermo Fisher Scientific, Waltham, MA, USA), sectioning at 3 μm (Epredia™ HM 325 microtome, Waltham, MA, USA), and subsequent hematoxylin-eosin staining (Gemini AS Automated Slide Stainer, Thermo Fisher Scientific) prior to microscopic examination.

### 2.3. Toxicological Analysis

For the determination of ethylene glycol, liver tissue samples were subjected to extraction with acetonitrile, followed by solid-phase purification using hydrophilic-lipophilic balance (HLB) cartridges. The extract was then derivatized with heptafluorobutyric anhydride and subsequently analyzed by gas chromatography mass spectrometry [55].

### 2.4. Statistical Analysis

A comparison of the present study with that of Rebollada-Merino et al. (2020) [32] was performed using a Chi-square test of independence (χ^2^) on the total numbers of non-natural and natural deaths in each study. The statistical analysis was conducted using GraphPad Prism 8 (La Jolla, San Diego, CA, USA), with significance set at *p* < 0.05.

### 2.5. Determination of Cause, Manner and Mechanism of Death

Suspected abuse referred to cases in which intentional harm to the animal was presumed by the submitters (law enforcement agencies, colony caretakers, or cat owners) who sent cats to the diagnostic center, representing either the direct cause of death or playing a contributory role in the fatal outcome. Cause, manner and mechanism of death were established based on gross and histopathological findings. Cause of death was defined as the main lesion leading to death, manner of death was classified as natural, non-natural, or undetermined, according to a forensic approach adapted to veterinary medicine [56,57,58]; and the mechanism of death was defined as the physiological derangement that resulted in the death [17]. The classification was performed retrospectively, after all potential causes of death in the analyzed cases had been determined and quantified. Non-natural deaths were grouped into the following categories: blunt force trauma, projectile injury and poisoning. Natural deaths were also categorized and reflected in different categories: infectious disease, renal disease, neoplasia, cardiac disease and anaphylactic shock. The mechanisms of death in non-natural cases were classified as hypovolemic shock, neurogenic shock, and acute renal failure.

## 3. Results

### 3.1. Case Distribution over Time

Between 2020 and 2024, 53 cats underwent forensic postmortem examination. The year with the highest number of submissions was 2023, with 18 cases, followed by 12 cases in 2020, 11 in 2024, seven in 2021, and five in 2022.

### 3.2. Animal Identification

Only 13 of the animals were identified through their microchip number (24.52%). Most of the cats were adults (*n* = 37), followed by juveniles (*n* = 10) and kittens (*n* = 6). The most common breed was the European shorthair (*n* = 48), followed by the Siamese (*n* = 2), British shorthair (*n* = 1), Sphinx (*n* = 1), and Ragdoll (*n* = 1). Regarding sex and carcass preservation, 30 (56.60%) of the animals were female and 23 (43.39%) were male, while 42 (79.24%) were classified as fresh, seven (13.20%) as advanced decay, and four (7.54%) as bloat. Upon receipt at our center, 25 carcasses were maintained under refrigeration (47.16%), while the remaining 28 were stored under freezing conditions (52.83%).

### 3.3. Manner, Cause, and Mechanism of Death

A greater number of cases involved non-natural causes of death (*n* = 31, 58.49%) compared to natural causes (*n* = 21, 39.62%). Among the non-natural deaths, 17 cases of blunt force trauma (32.07%), eight poisonings (15.09%) and six projectile injuries (11.32%) were described. Regarding natural causes of death, 15 cases involved infectious diseases (28.30%), three renal diseases (5.66%), and one case each of anaphylactic shock (1.88%), cardiac disease (1.88%), and neoplasm (1.88%). The cause of death remained undetermined in 1.88% (*n* = 1) of the cases as the cause of death could not be specified.

The main gross lesions associated with blunt force trauma were subcutaneous hematomas varying in extent and severity between animals. These were most frequently observed in the head, cervical region, both hemithoraces, and the abdominal region, often associated with lacerations and contralateral abrasions. Both bilateral epistaxis (Figure 1A) and traumatic abdominal herniations were commonly observed as consequences of high-impact trauma. The herniations were secondary to rupture of the oblique, transverse, and/or rectus abdominal muscles, often resulting in external exposure of the abdominal viscera (Figure 1B) and associated with extensive hematomas. Regarding the abdominal viscera, several cases presented hepatic, spleen and/or kidney rupture caused by the kinetic energy of the trauma (Figure 1C,D), associated with severe hemoperitoneum. In the thoracic cavity, total or partial rib fractures were observed in several cases, together with hemothorax (Figure 1E), multifocal to coalescing pulmonary hemorrhages, focal or multifocal subpleural hematomas (Figure 1F), and hemopericardium secondary to cardiac injuries. Aortic rupture resulting from high-impact thoracic trauma, with consequent severe hemopericardium, was observed in a single animal (Figure 2A). Multiple fractures of both long and short bones were identified, particularly affecting the mandible (Figure 2B), limbs, and vertebral column. Head lesions primarily consisted of skull fractures with submeningeal hemorrhages. The presence of hyphema along with episcleral hemorrhages (Figure 2B) was frequent, and often associated with subcutaneous periorbital hematomas. Some animals displayed partial or complete splintering of the nails of one or more limbs. Secondary lesions attributable to hypovolemic shock, such as pale mucous membranes and exsanguinated spleen were also present.

Regarding projectile injuries, lesions were mainly located on the thorax, head, and forelimbs. The most affected organs were the lungs, heart, and brain. Subcutaneous and muscular hemorrhages of variable extension were observed surrounding the entry and exit wounds, corresponding to ballistic trajectories (Figure 2C). Based on the macroscopic characteristics of each lesion, including entry and exit wounds, intensity and extent of the injuries, and the projectiles and/or fragments recovered during necropsy, it was determined that all six projectile-related injuries were consistent with air-powered weapons, with a retained pellet identified within the affected tissues in one of the cases (Figure 2D).

Histopathologically, both causes of non-natural deaths showed pronounced erythrocyte extravasation in the affected regions (Figure 2E), indicative of significant vascular disruption and tissue injury, particularly in the dermis, skeletal muscle, lungs and liver. Large numbers of macrophages showed erythrophagocytosis, frequently associated with blood resorption in regional lymph nodes. In skeletal muscle, trauma-induced lesions included Zenker degeneration, characterized by hypereosinophilia, loss of striation and nuclear pyknosis, often progressing to necrosis and accompanied by a marked neutrophilic inflammatory response. The main final mechanism of death for both non-natural deaths was hypovolemic shock, along with neurogenic shock secondary to severe non missile and missile brain injuries.

A total of eight cases (15.09%) were diagnosed with ethylene glycol poisoning, confirmed through toxicological analysis. Macroscopic examination showed non-specific lesions, including hydrothorax, hydropericardium, and vascular renal ectasias. Histopathologically, calcium oxalate crystals were observed as irregular, refringent and fragmented formations predominantly located within the proximal convoluted tubules of the kidney (Figure 2F) and intravascularly in the brain and lungs, associated with acute tubular necrosis. The final mechanism of death for this cases was acute kidney failure. Table 1 summarizes the principal findings associated with each non-natural cause of death documented in this study.

Among natural causes of death, a significant proportion of cats were submitted with bacterial and mycotic bronchopneumonia (47.61%), often accompanied by lymphoid tissue atrophy with lymphocytolysis, as well as lymphoplasmacytic rhinitis and tracheitis, suggesting concomitant primary viral infections. Additionally, fibrinous polyserositis (Figure 3A) and perivascular pyogranulomatous inflammation (Figure 3B) were recurrently observed, strongly indicative of feline infectious peritonitis virus. Both macroscopic and histopathological lesions consistent with chronic renal disease were identified, with one case demonstrating secondary cardiac remodeling (Figure 3C) and associated systemic vascular thrombosis (Figure 3D). Finally, gastrointestinal and pulmonary parasitism by nematodes and cestodes were also a common finding.

The Chi-square test (χ^2^) revealed a significant difference in the distribution of natural and non-natural outcomes between the two studies (*p* = 0.0191).

### 3.4. Relationship Between Suspected Abuse, Manner, and the Cause of Death

Poisoning was the most prevalent category of suspected abuse (*n* = 31; 58.49%), followed by undetermined abuse (*n* = 16; 30.18%), projectile injury (*n* = 3; 5.66%), and blunt force trauma (*n* = 3; 5.66%).

Based on definitive diagnoses, of the 31 animals suspected of poisoning, 17 (54.83%) were confirmed as non-natural deaths (ten blunt force trauma and seven poisonings), while 13 (41.93%) were classified as natural deaths (eleven infectious diseases, one renal disease, and one neoplasm). The cause of death in one case could not be specified due to the advanced state of decomposition.

Regarding the 16 cases of suspected unspecified abuse, eight (50%) were grouped as non-natural deaths (three projectile injuries, four blunt force traumas, and one poisoning), and eight (50%) as natural deaths (four infectious diseases, two renal diseases, one cardiac disease, and one anaphylaxis).

In all cases of suspected projectile injury (*n* = 3) and blunt force trauma (*n* = 3), the diagnosed cause of death coincided with the suspected abuse. Table 2 summarizes the relationship between suspected animal abuse and the final forensic diagnosis.

## 4. Discussion

Throughout the five-year study period, a total of 53 cases were referred on suspicion of feline abuse, of which 31 (58.49%) were determined to have died from non-natural causes. The highest number of submissions occurred in 2023. In contrast to the 2014–2019 study [32], a total of 41 cats with suspected animal abuse were diagnosed with 14 (34.14%) non-natural deaths, suggesting a possible upward trend in the number of cats referred with suspected animal abuse (53 cases over a 5-year period, versus 41 cases over a 6-year period). This observed trend, both within Spain and globally [2,12,51,59], is likely attributable to increased public awareness and sensitivity towards animal welfare.

Of all cats diagnosed, 40 (75.47%) were not microchipped, consistent with the findings of Rebollada-Merino et al. (2020) [32]. According to the Spanish Law 7/2023 of 28 March [44], it is the owner’s responsibility to ensure that their cats are microchipped for identification purposes (except for cats under six months of age). In Spain, microchipping enables identification of the animal and its owner, as well as the veterinarian who has administered the chip, thereby facilitating the prosecution and punishment of individuals responsible for animal abuse.

Stray cats are particularly vulnerable since they are not registered in any database, being ideal targets for those with criminal intent. This highlights the fundamental principle that reliable identification of all animals within a given region is indispensable for the effective investigation and prosecution of abuse cases. Beyond individual recognition, systematic identification allows accurate quantification of the animal population, which is crucial for monitoring, surveillance, and the development of preventive strategies. In the absence of such measures, systematic case follow-up is severely hindered, and animal welfare may ultimately be compromised. The current low rate of microchip identification among the feline population represents a significant challenge to be addressed in Spain, particularly in the context of forensic investigations and responsible ownership.

Regarding age, adult cats represented the predominant age group in this study, with proportions of this group exceeding those reported in other studies [12,39]. The proportion of males and females in both studies was similar, which also coincides with other studies internationally [12,39,59].

In contrast to the findings reported by Rebollada-Merino et al. (2020) [32], the number of non-natural deaths observed in the present study (*n* = 31) exceeded that of natural deaths (*n* = 21). This increase should be investigated in future studies to predict their trend and impact. Specifically, blunt force trauma was the most frequently diagnosed cause of non-natural death, consistent with global trends [12,39,59]. It is essential to adequately distinguish between different types of traumatic lesions and consistently rely on histopathological analysis to accurately assess its chronicity, relating these findings to events that contribute to the spatiotemporal epidemiological reconstruction of each case [60].

Moreover, distinguishing antemortem from postmortem lesions remains a significant challenge. Forensic pathologists must therefore possess adequate training to identify vital reactions, including inflammatory responses, Zenker’s degeneration of muscle fibers, and intravital hemorrhages [60]. The lesion patterns observed in many of the cats with traumatic lesions in the present study are consistent with previously reported findings [12,39,59,60]. The low plasticity of abdominal parenchymal organs, notably the liver, spleen, and kidneys [60], likely accounts for the high incidence of organ ruptures observed in the present study. In addition, the presence of partial or complete fractures of nails may reflect attempts by the animals to stabilize themselves during significant traumatic displacement and should be regarded as a noteworthy necropsy finding by veterinary forensic pathologists, as they are commonly found in cats that are hit by motor vehicles [58].

Macroscopic injuries produced by projectiles were similar in both studies, although their incidence was lower in 2020 [32]. The presence of retained projectiles was evidenced in one of the six cases. In Spain, access to firearms is highly restricted, whereas air-powered weapons are relatively common. This may explain why all projectile-induced injury cases in the present study involved air guns, since no arm license is required [61]. These weapons have a lower capacity to inflict tissue damage because of the reduced kinetic energy transmitted to the projectile [58]. Sociocultural factors specific to each country, as well as the victim’s epidemiological context, should also be carefully considered.

Histopathological examination of the wound trajectory margins is essential to detect gunpowder residues and assess tissue coagulation, which can confirm the use of a firearm and, in some instances, even allow for estimation of the shooting distance [62]. Special attention should be paid to the entry and exit trajectories of projectiles, along with the resultant lesions sustained in the organs [62]. Each projectile-related injury event must be considered as a distinct and individual case [63]. Firearm injuries exhibit a significantly higher prevalence in other regions of the world, particularly in the United States [62], with notable concentrations in rural and peri-urban areas [64]. In such contexts, the use of radiographic imaging prior to macroscopic examination is highly valuable for accurately determining the spatial location of the projectile or its fragments [62].

Another notable difference from the study published in 2020 [32] is the new identification of poisoning as a non-natural cause of death in the present study. All confirmed poisoning cases (*n* = 8) involved ethylene glycol intoxication. This finding contrasts with the results obtained from other international studies, in which poisonings caused by carbamates and various rodenticides were more frequently reported [34,35,36,37,38,39,65,66,67]. Nevertheless, the overall incidence of confirmed poisoning cases was markedly low in comparison to the prevalence reported in the aforementioned literature [26,34,36,37,38,39,65].

The occurrence of poisoning cases in feline colony environments warrants careful consideration. A thorough assessment is required to differentiate accidental intoxication from deliberate poisoning. Identification of the toxic agent can assist forensic pathologists in determining intentionality of the poisoning [58]. In Spain, carbamates are prohibited for use outside agricultural contexts, whereas substances such as ethylene glycol and rodenticides remain legally available and are commonly encountered in urban environments, where they serve as automotive antifreeze and rodent control agents, respectively [35,67]. It should be noted that intentional poisoning may be mistakenly assumed when animals are, in fact, exposed to environmental toxic agents [35]. Consequently, laboratory analysis of bait constitutes a tool as important as molecular detection in the organs themselves, as it enables the epidemiological investigation of poisoning events [65,67].

Consistent with the present findings, the literature indicates that ethylene glycol intoxication shows unspecific macroscopic lesions, including pleural and peritoneal effusions, pulmonary congestion and edema, hepatic congestion, pale kidneys, and nephromegaly [68]. These observations underscore the critical role of histopathological examination as an essential diagnostic tool in forensic investigations.

In both studies, every case was categorized as a single type of non-natural death, with no co-occurrence of different categories. Moreover, the lesions did not exhibit definitive indicators of chronicity, suggesting that, if the animals had been subjected to abuse, it was not prolonged or sustained. Repetitive injuries as a consequence of chronic animal abuse are of great importance in other publications [69] and should be given special attention. No evidence of sexual animal abuse was found, although this type of abuse is a real problem in Spain, other nearby countries [70], and worldwide [39,69,71]. Other non-natural findings such as thermal or electrical injuries [72,73], bite injuries [12,32], signs of drowning [27,74,75] or starving were not found, the last one being frequently reported in dogs [76].

Infectious diseases hold significant relevance in the present study, as well as in the one published in 2020 [32], and represent the leading cause of natural deaths. The high incidence of infectious diseases found in both studies coincides with other reports focused on feline colonies [77]. The increasing social awareness, fostered over time by the development of veterinary forensics, has contributed to a higher number of suspected cases being submitted for investigation. This, in turn, may lead to an overrepresentation of certain diseases, particularly among stray and colony cats, which appear to be the most susceptible populations.

Suspicions of animal abuse reported upon submission demonstrated minimal concordance with the final diagnoses, mirroring the pattern reported by Rebollada-Merino et al. (2020) [32], in which the incidence of suspected poisoning was also considerable. In light of this, pathologists must exercise caution when considering the initial suspected cause of death and prioritize diagnoses based on objective macroscopic and histopathological findings. Reliance on the reported history of the animal is often inaccurate, especially in cases of severe neglect [1], where critical information may be incomplete or inaccurate. It is worth noting that the high number of cases submitted with unspecified suspicions of abuse (*n* = 16; 30.18%) is likely attributable to increased societal awareness regarding animal welfare. This trend is highly positive, as it encourages the forensic investigation of animals that have died under unclear circumstances, even when no specific form of abuse is initially suspected, thereby enhancing the prosecution of crimes against animal welfare and potentially contributing to the prevention of interpersonal violence.

It is important to emphasize that most of the cats were brought from stray cat colonies (90.56%). In Spain, law 7/2023 of March 28 [44], properly covers the concepts of feline colonies, as well as their needs and obligations of those responsible for them. This law mandates the microchipping of cats within colonies, as well as the implementation of an appropriate health management program, including deworming and sterilization. In both studies [32], the very low number of microchipped cats reported, and the high prevalence of infectious diseases, in addition to gastrointestinal and pulmonary parasites, suggests that the conditions of these animals are often inadequate, which predisposes them to stress and resultant immunosuppression. Neglect is interpreted as cases where necessities of life (food, water, and shelter) or appropriate veterinary care was not provided, leading to chronic injuries, weakening, or death [58]. In this study, animals from colonies with poor health and welfare conditions that ended up dying from infectious diseases were classified as natural deaths, although based on the definition of neglect, the actual significance of these cases should be re-evaluated, and their complex classification should be further investigated.

Despite the substantial development of veterinary forensics internationally [19,25,26,78], evolution remains highly heterogeneous, with significant variability in legislation, specialized forensic training [25,26] and territorial scientific bibliography [8]. This heterogeneity hinders a clear understanding of the actual state of animal welfare and prevents effective collaboration between countries. Animal welfare should be a priority for contemporary society, especially considering its association with interpersonal violence [1,3,4,5,6]. Specific laws have been created to protect animals worldwide; however, in societies with high rates of violence, animals are common targets of abuse, even where animal welfare laws exist [39]. If animal welfare is not protected globally, veterinary forensics will not be able to grow to its full potential, making collaboration between human medicine and veterinary forensics [11,21] essential, especially in countries with limited resources dedicated to the One Welfare concept [9].

It is important to emphasize that some limitations are inherent to this retrospective study. Although all cats included were submitted under genuine suspicion of animal abuse, relying solely on macroscopic and histopathological examinations makes it challenging to discern which of the non-natural deaths were the result of physical abuse or accidental injuries, since interpretation of intentionality based solely on lesions observed in a carcass is extremely complex, and its precise assessment during necropsy is inherently limited. In order to reliably distinguish between these conditions, greater collaboration between law enforcement agencies and diagnostic laboratories is required, together with a complete evaluation of the crime scene, both of which were very limited in our study. Therefore, the assumption that all non-natural deaths assessed in this study were attributable to animal abuse cannot be made, particularly in cases of blunt force trauma, where many incidents could be linked to motor vehicle accidents. Although some studies have attempted to elucidate the main differences between accidental and non-accidental blunt force trauma [79], this distinction continues to represent a major diagnostic challenge. By contrast, in relation to projectile injuries, the likelihood of accidents involving such weapons in Spain is relatively low, which strongly supports their association with potential animal abuse.

Although the diagnostic protocol considered the possibility of detecting other toxic agents, the absence of lesions suggestive of their involvement resulted in the omission of further toxicological testing beyond ethylene glycol. Consequently, the complete absence of other toxic molecules in some animals cannot be excluded.

Finally, even though the data suggest an upward trend in cases suspected of animal abuse referred to the diagnostic center, the comparative analysis was descriptive in nature and limited to different annual periods. Therefore, long-term statistical studies are required to more accurately characterize the evolution of this trend over time.

In the context of Spain, the present study constitutes the second investigation aimed at assessing the prevalence of animal abuse within the Community of Madrid and represents the first research effort specifically focused on feline populations. Since the publication of the study carried out by Rebollada-Merino et al. (2020) [32], there has been an increase in the number of cases involving suspected animal abuse submitted within this region. This trend underscores the potential societal impact of such research in raising awareness among veterinary and non-veterinary professionals, as well as society, regarding the manifestations and forensic relevance of animal abuse. These findings suggest that scientific contributions in this domain may play a critical role in enhancing social consciousness and responsiveness to animal welfare issues. Nonetheless, persistent structural limitations such as inadequate attendance at crime scenes, and insufficient interinstitutional coordination between state authorities and veterinary forensic diagnostic laboratories, continue to constitute significant barriers to the advancement and institutionalization of veterinary forensics in Spain.

## 5. Conclusions

In the present study, the data suggest a possible upward trend in the number of cats referred with suspected animal abuse, likely attributable to an increased public awareness and sensitivity toward animal welfare issues. Furthermore, the emergence of new cases of poisoning in this region, together with the poor correlation between initial suspicion of animal violence and the final diagnosis, should be taken into consideration when assessing future cases. Cat colonies continue to be a point of societal conflict, and forensic findings suggest that they might be a primary target for this type of abuse. Despite the ongoing expansion of veterinary forensics in Spain, further research across diverse geographical regions of the country is required to obtain a more accurate estimate of the true prevalence of this condition.

## Figures and Tables

**Figure 1 animals-15-02892-f001:**
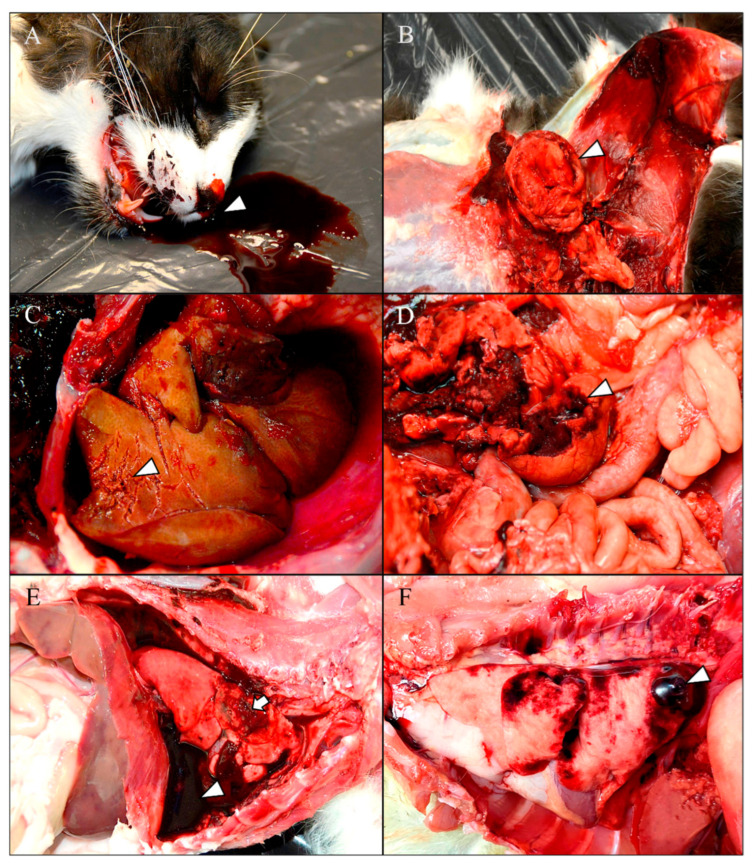
Macroscopic findings in cats with suspected feline abuse. (**A**) Severe bilateral epistaxis (arrowhead) associated with pale gingival mucosa. (**B**) Evisceration of the small intestine (arrowhead) secondary to high-impact abdominal trauma, associated with a focally extensive hematoma in the left hind limb. (**C**) Severe hepatic rupture (arrowhead) with secondary hemoperitoneum. (**D**) Severe renal rupture (arrowhead) secondary to high-impact abdominal trauma. (**E**) Severe hemothorax (arrowhead) associated with rupture of the middle right lung lobe (arrow). (**F**) Dorsocaudal subpleural hematoma in the left caudal lobe (arrowhead) associated with multiple intraparenchymal hemorrhages.

**Figure 2 animals-15-02892-f002:**
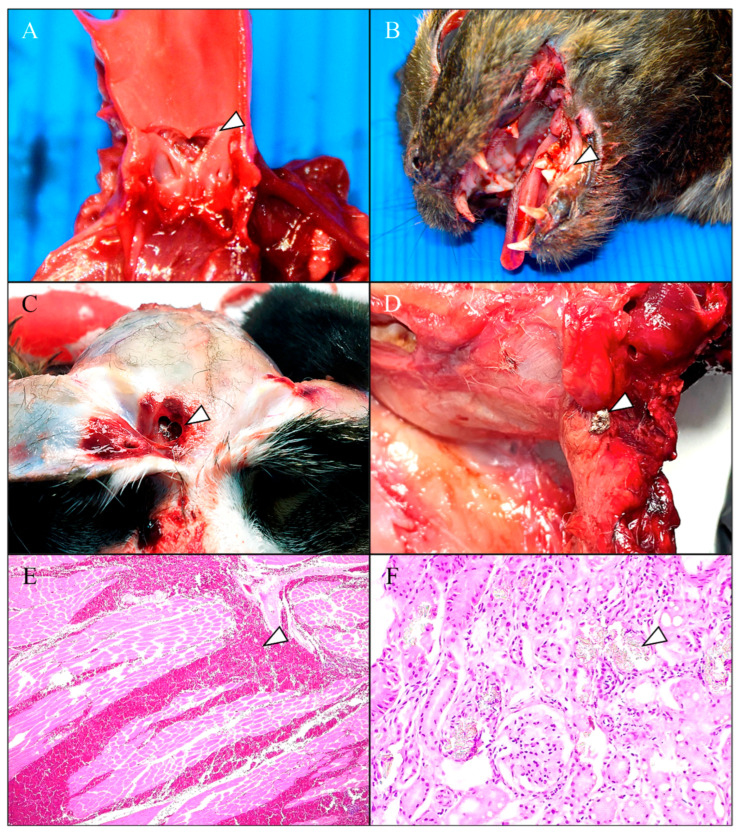
Macroscopic and histopathologic findings in cats with suspected feline abuse. (**A**) Supravalvular aortic rupture (arrowhead) secondary to high-impact thoracic trauma. (**B**) Left unilateral mandibular fracture (arrowhead) associated with severe episcleral hemorrhage secondary to high-impact trauma. (**C**) Projectile injury to the frontal bone (arrowhead). (**D**) Remaining intracranial projectile (arrowhead) with multiple encephalic hemorrhages. (**E**) Severe multifocal muscular hemorrhages (arrowhead). HE, ×40. (**F**) Acute tubular necrosis associated with the presence of intratubular calcium oxalate crystals (arrowhead). HE, ×200.

**Figure 3 animals-15-02892-f003:**
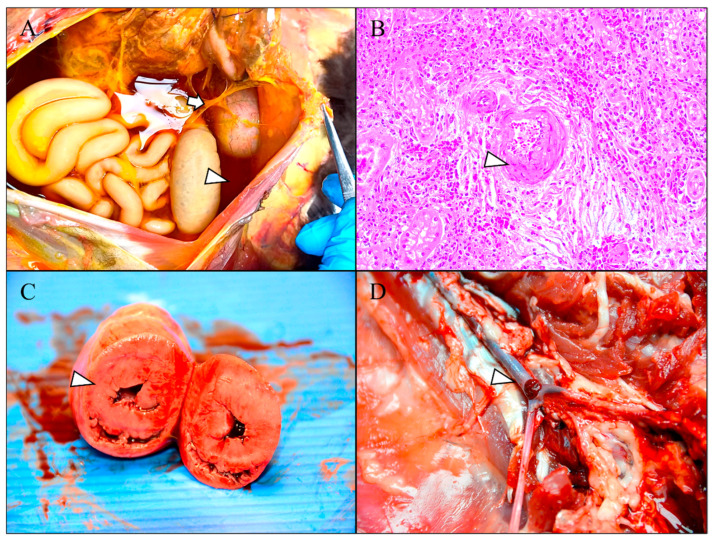
Macroscopic and histopathologic findings in cats with suspected feline abuse. (**A**) Severe fibrinous peritonitis (arrow) associated with hydroperitoneum (arrowhead) compatible with a feline infectious peritonitis virus infection. (**B**) Pyogranulomatous vasculitis associated with moderate vacuolization of the tunica media (arrowhead) and renal tubular degeneration compatible with a feline infectious peritonitis virus. HE, 200×. (**C**) Concentric myocardial hypertrophy of the left ventricle with intense reduction in its lumen (arrowhead). (**D**) Thrombosis at the bifurcation of the external iliac arteries (arrowhead) due to hypertrophic cardiomyopathy of the left ventricle.

**Table 1 animals-15-02892-t001:** Main macroscopic and histopathological findings in cats suspected of animal abuse.

Type of Non-Natural Death	Macroscopic Findings	Histopathological Findings
Blunt force trauma (*n* = 17)	Subcutaneous hematomas (head, thorax and abdomen) Mandible, limbs, ribs and skull fractures Traumatic abdominal herniations Hepatic, spleen and kidney rupture Hemoperitoneum, hemothorax and hemopericardium Hyphema, episcleral and periorbital hemorrhages Splintered nails	Extensive erythrocyte extravasation (skin, skeletal muscle and viscera) Lymph node blood resorption Zenker degeneration Erythrophagocytosis
Poisoning (*n* = 8)	Hydrothorax and hydropericardium Vascular renal ectasias	Intratubular and intravascular calcium oxalate crystals (kidney tubules, lungs and brain) Acute tubular necrosis
Projectile Injuries (*n* = 6)	Entry and exit ballistic trajectories Subcutaneous hemorrhages around trajectories Hemothorax and hemopericardium Pulmonary hemorrhages Skull fractures	Extensive erythrocyte extravasation (skin, skeletal muscle, viscera and brain) Lymph node blood resorption Zenker degeneration Erythrophagocytosis

**Table 2 animals-15-02892-t002:** Relation between initial suspected abuse and diagnosed cause of death.

Cause of Death Suspected by Submitters	Cause of Death Diagnosed by the Forensic Necropsy
Poisoning (*n* = 31)	Infectious diseases (*n* = 11) Blunt force trauma (*n* = 10) Poisoning (*n* = 7) Renal disease (*n* = 1) Neoplasms (*n* = 1)
Unspecified abuse (*n* = 16)	Blunt force trauma (*n* = 4) Infectious diseases (*n* = 4) Projectile injury (*n* = 3) Renal disease (*n* = 2) Poisoning (*n* = 1) Cardiac diseases (*n* = 1) Anaphylaxis (*n* = 1)
Projectile injury (*n* = 3)	Projectile injury (*n* = 3)
Blunt force trauma (*n* = 3)	Blunt force trauma (*n* = 3)

## Data Availability

All data are contained within this manuscript.

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
