# Peer review of "Evolution of Suspected Cat Abuse Between 2020 and 2024 in the Community of Madrid (Spain)"

_animals, 2025, doi:10.3390/ani15192892_

Round 1
Reviewer 1 Report
Comments and Suggestions for Authors
Thank you for the opportunity to review this paper. The topic of the prevalence of animal cruelty is one on which we lack data, both in Spain and globally, so I welcome any addition to the literature.
Having said this, the paper is in need of a good deal of additional work before it is ready for publication.
- Throughout the paper, clarify the use of “suspected abuse”: does this refer to the initial suspicion of the referring agency (e.g., law enforcement), or to the pathologist’s findings? This distinction is particularly important given the lack of agreement between the initial suspected cause of death and the exam results.
- The terms “non-natural death” and “abuse” are occasionally used interchangeably or in proximity to one another, which causes confusion because non-natural death can be accidental (and thus is not always due to abuse). For example, an animal accidentally hit by a car. l. 381-382 classify electrical injuries and drowning as nonaccidental, but these could also occur by accident. Truly only the projectile injuries can be reasonably assumed to be more likely due to abuse than accident. The paper would benefit from refocusing on non-natural death as the outcome variable, highlighting the distinction between non-natural vs. nonaccidental (abuse), and emphasizing more strongly that of the cases diagnosed as non-natural deaths, no determination was made that these were due to intentional abuse (which should be discussed in a Limitations section. Presenting the legal definition of abuse in Spain would also be helpful.
- Re: blunt force trauma, did the authors reference the research on differentiating accidental from nonaccidental BFT?—e.g., Intarapanich et al.
- I am unclear how the authors’ key conclusion that “there was an increase of nearly four suspected abused cats per year” was calculated. Does this refer to the difference between 14 cats in the prior study having been diagnosed with non-natural deaths vs. 31 in the current study? If so, this is a problem because a) again, suspected abuse and non-natural death are not the same thing and b) when looking at numbers by year, there is no evidence of an upward year-over-year trend, as is implied by the statement. Further, raw numbers are relevant only in relation to their denominators: that is, the number of cases of interest relative to the overall number of cases. If for example the total number of cats submitted to the center had skyrocketed to 100 during the time period of the present study, then the finding of 31 cases of non-natural death would be roughly similar to the previous finding of 14 cases of non-natural death among 41 presented animals. Accordingly, it would be strongly preferable to present the comparison between studies as a rate (percentage): 14/41 vs. 31/53.
- 54-55 Not all deceased animal victims of suspected abuse are examined by a pathologist. Some are examined by a forensic veterinarian. It would be helpful in this paragraph to add a bit more about the various veterinarians who may assist in abuse cases (pathologist, forensic vet, general vet) and how they differ.
- 98-102 it seems possible (even likely) that the reverse causal pattern is actually the case: increase in societal awareness/concern driving legislation and scholarly interest.
- 185-186 If intended to create generalizable knowledge (that is, to be interpreted outside this study and this sample), this result warrants a test statistic (z-test of proportions) to establish whether the observed difference is statistically significant.
- 220 This phrase is unclear in the context of the sentence: “associated with social and legislative framework of Spain”
- 303-305 It is unclear how microchipping facilitates prosecution of animal abusers unless the owner is also the abuser.
- 310 “these being slightly higher” –this is unclear—does this mean that the proportion of adult cats in this study was higher than the proportion of adult cats in the other studies cited?
- 311-312 “The number of males and females in both studies was similar”—does this mean the number was similar across studies (# of males in study 1 similar to # of males in study 2, same for females), or that the proportion of males vs. females in each study was similar, or that the number of males and females was similar within each study?
- 358-366 Re: the ethylene glycol poisonings, did these cases originate from the same area? How do we know these poisonings were deliberate and not the result of accidental exposure? This is discussed later in the paper but is potentially misleading here, and is conflated elsewhere e.g. in the Simple Summary, l. 20: “suspected abuse cases increased, including new anti-20 freeze poisonings.”
- In l. 425, I would encourage rewording: the relationship between animal welfare and interpersonal violence is not necessarily one of causal “impact” (e.g., in the two studies referenced here that focus on domestic violence, animal abuse does not cause the interpersonal violence, but instead is associated with the interpersonal violence as part of the overall dynamic of violence).
- The paper requires a Limitations section.
- Was this research reviewed/overseen by an ethics committee tasked with animal care and use?
- The paper needs work prior to resubmission to correct errors in the use of English: for example, l. 46 “with” instead of “being”; l. 59 “suffered” is used incorrectly in this context—something like “experienced” would work.
Reviewer 2 Report
Comments and Suggestions for Authors
Reviewer Comments / Questions:
- In this study, the authors broadly categorized manner of death into non-natural and natural deaths. Why did you not further distinguish between accidents and abuse as a subcategory of non-natural deaths? What was the author's basis for diagnosing cats that had died from non-natural and blunt force trauma as a result of abuse rather than an accident? Non-natural deaths include accidental deaths caused by cars or motorcycles, and accidental predation by wild animals. Were these possibilities taken into consideration in this study?
- The mechanisms of death were classified as blunt force trauma, firearm injury, and poisoning in this study, but why did the authors not consider the possibility of sharp force injury or asphyxiation?
- Why did you not determine the type of abuse in this study, such as physical abuse, neglect, or sexual abuse?
- Does this study include cats with missing body parts?
- Why did the authors investigate only ethylene glycol as a possible cause of poisoning, and no other poisons or drugs?
- The usefulness of postmortem imaging examination before necropsy was mentioned in the discussion section, but why was this not performed in this study?
- In the authors' necropsy cases of the cat colony, were there any cases where there was evidence of interpersonal violence?
Comments and Suggestions for Authors
This study investigated changes over time in the causes, manners, and mechanisms of death of cat colonies diagnosed as unnatural in a limited area, and noted an increase in the number of cats for which necropsies were requested due to suspected abuse. Furthermore, the study noted that the mechanisms of unnatural death suspected by requesters based on appearance and circumstances? were significantly different from those revealed by necropsy, qualitative ethylene glycol analysis, and histopathological examination. Therefore, this study points out the gap in judgment between veterinary pathology experts who are familiar with the knowledge of forensic veterinary medicine and non-experts, and may provide evidence to assert the importance of forensic veterinary medicine in veterinary medicine and the need for education in it. However, in this study, causes of death were broadly divided into natural and non-natural deaths, and blunt trauma was listed as the mechanism of death, but no scientific evidence was presented to rule out accident and determine that it was abuse. If the author wishes to state that this study is an investigation into cat abuse, they must provide scientific evidence distinguishing accidents from abuse. If It is difficult to modify your manuscript, this study have not be the investigation of animal abuse, but "that of the mechanism of deaths targeting colony of cats diagnosed as non-natural by a veterinary pathologist."
Title
Of the non-natural deaths, particularly for cats which mechanism of death were blunt trauma, the basis for the diagnosis of abuse or accidental is not stated.
Therefore, this study is the result of an investigation into the cause, type, and mechanism of death of a cat colony "suspected" of abuse, and the title should be changed to "Evolution of cat had been suspected abuse between 2020 and 2024 in the Community of Madrid (Spain)."
Simple Summary
Abstract
—Same comment regarding the mechanism of death in non-natural death and the differentiation of abuse or accidental.
Keywords
non-natural death can be added.
postmortem necropsy
Since "necropsy" is performed after death, delate "postmortem necropsy" and add "forensic necropsy."
Introduction
What about explaining the different types of abuse, such as physical abuse (non-accidental injury), neglect, and sexual abuse?
Please add an explanation as to why a clear diagnosis of abuse and the type of abuse could not be provided.
Please revise the purpose of this study to: "To clarify whether the manner of deaths was natural or non-natural, to diagnose the cause and mechanism of death, and to investigate these changes of over time."
Line 114
These types of studies
Materials and Methods
2.1. Cases submitted and postmortem procedure
Lines 124-126
Do you have any detailed information about the person had been suspecting abuse of cat? For example, how about categorizing and adding for persons, had been suspecting abuse of cat, such as the general public (passersby), clinical veterinarians, police, and animal administrative officer?
In the method section, please add the storage conditions of the bodies of cat until necropsy (such as frozen, refrigerated, and room temperature) and the days from discovery of the body until necropsy.
Was there any evidence that authors were experts of veterinary forensics?
Please explain the specific steps that make forensic veterinary necropsy different from pathological necropsy.
2.3. Toxicological analysis
If the ethylene glycol test was a quantitative test, please add the quantitative result to the results. Include the analytical equipment, manufacturer, region, and country.
Previous literature on animal cruelty has reported malicious poisoning using poisoned bait laced with pesticides. If you have not conducted toxicological analysis for poisons other than ethylene glycol, such as pesticides, please state the reason why you were unable to do so as a limitation of your research and add a note in the discussion section to encourage future research.
2.4. Determination of cause, manner and mechanism of death
Non-natural death should be further classified as accident or abuse.
The mechanisms of death should be classified as blunt trauma, firearm injury, poisoning, as well as sharp force injury, asphyxiation, and thermal injury.
Results
3.2. Animal identification
Is it common in Spain for cats in Madrid's cat colony to contain purebreds, such as e Siamese, British shorthair, Sphinx, and Ragdoll?
What is the reason for this?
3.3. Manner, cause, and mechanism of death
Please create a new table summarizing the findings and test results that support the definitive diagnosis of each cause of death.
3.4. Relationship between suspected abuse, manner, and cause of death
The mechanism of death is incorrectly listed as the type of abuse, which is confusing.
Please include in the methods section how the type of abuse was diagnosed.
If it is not possible to obtain results that allow for distinguishing between accidents and abuse, then the possibility of an accident cannot be ruled out as a final conclusion, and this should be stated in the discussion section as a limitation of the study.
Figure 1
Figure 2
Figure 3
You have attached necropsy photos and histopathological images of the cat had been suspected to be abused, but please also add the final diagnosis of whether or not the cat was abused, and the type of abuse (physical abuse).
Table 1.
The mechanism of death is incorrectly listed as the type of abuse, which is confusing. Please revise this to "the mechanism of death in the cat determined by the requester" instead of "Type of suspected abuse," and to "the mechanism of death in the cat diagnosed by the forensic necropsy" instead of "Final forensic diagnosis."
Discussion
—Same comment regarding the mechanism of death in non-natural death and the differentiation of abuse or accidental.
Since neglect was not the subject of this study, we propose to describe it as a limitation.
Line 375
Do you mean a single injury or ingestion of a poison? Please change it to "single injury or ingestion of a poisonous substance," as this could be an accident as well as abuse.
Round 2
Reviewer 2 Report
Comments and Suggestions for Authors
I think that the manuscript has been sufficiently improved to warrant publication in Animals.